# Supramolecular double-stranded Archimedean spirals and concentric toroids

Norihiko Sasaki [1,2], Mathijs F. J. Mabesoone [3], Jun Kikkawa[2], Tomoya Fukui [2], Nobutaka Shioya[4], Takafumi Shimoaka [4], Takeshi Hasegawa [4], Hideaki Takagi[5], Rie Haruki[5], Nobutaka Shimizu [5], Shin-ichi Adachi [5], E. W. Meijer [3], Masayuki Takeuchi [2✉] & Kazunori Sugiyasu [1,2✉]

Connecting molecular-level phenomena to larger scales and, ultimately, to sophisticated molecular systems that resemble living systems remains a considerable challenge in supramolecular chemistry. To this end, molecular self-assembly at higher hierarchical levels has to be understood and controlled. Here, we report unusual self-assembled structures formed from a simple porphyrin derivative. Unexpectedly, this formed a one-dimensional (1D) supramolecular polymer that coiled to give an Archimedean spiral. Our analysis of the supramolecular polymerization by using mass-balance models suggested that the Archimedean spiral is formed at high concentrations of the monomer, whereas other aggregation types might form at low concentrations. Gratifyingly, we discovered that our porphyrin-based monomer formed supramolecular concentric toroids at low concentrations. Moreover, a mechanistic insight into the self-assembly process permitted a controlled synthesis of these concentric toroids. This study both illustrates the richness of self-assembled structures at higher levels of hierarchy and demonstrates a topological effect in noncovalent synthesis.

[1] Department of Materials Physics and Chemistry, Graduate School of Engineering, Kyushu University, 744 Moto-oka, Nishi-ku, Fukuoka 819-0395, Japan. [2] National Institute for Materials Science, 1-2-1 Sengen, Tsukuba, Ibaraki 305-0047, Japan. [3] Laboratory of Macromolecular and Organic Chemistry and the Institute for Complex Molecular Systems, Eindhoven University of Technology, P.O. Box 513, Eindhoven 5600 MB, The Netherlands. [4] Laboratory of Chemistry for Functionalized Surfaces, Division of Environmental Chemistry, Institute for Chemical Research, Kyoto University, Gokasho, Uji, Kyoto 611-0011, Japan. [5] Photon Factory, Institute of Materials Structure Science, High Energy Accelerator Research Organization, Tsukuba, Ibaraki 305-0801, Japan. ✉email: TAKEUCHI.Masayuki@nims.go.jp; SUGIYASU.Kazunori@nims.go.jp

Although the principles of molecular self-assembly have become well understood through the development of supramolecular chemistry[1–4], unexpected and intriguing self-assembled structures continue to be discovered, the diversity of which increases as the number of molecular components increases. For example, Fujita et al.[5] discovered a new polyhedron, the existence of which was not predictable from their previous studies. Once a graph theory was established for the new family of polyhedrons, it was possible to predict the formation of a larger member in the family, which was later experimentally confirmed. One of us[6] identified pathway complexity in supramolecular polymerization, the mechanism of which was elucidated experimentally and computationally, thereby permitting the synthesis of an otherwise thermodynamically inaccessible helical supramolecular polymer. In such cases, weak interactions and subtle structural constraints accumulate, and kinetic effects often play significant roles, resulting in unexpected outcomes. It thus appears that principles of molecular self-assembly established for small discrete structures, such as host–guest systems or interlocked molecules, cannot simply be extended to design much larger structures. Consequently, molecular self-assembly at a higher hierarchical level remains largely unexplored[7–13].

Here we report two unusual self-assembled structures formed from a porphyrinato zinc derivative bearing fluorinated side chains, **6FF** (Fig. 1a). Previously, we found that an analogous porphyrin **6HH** forms one-molecule-thick two-dimensional (2D) supramolecular nanosheets (Figs. 1b and 2b)[14]. In these nanosheets, porphyrin cores of **6HH** stack in a short-slipping J-aggregation mode, as characterized by a split Soret band (Fig. 2a), and the 2D growth is driven by van der Waals forces between the hexyl chains (see the $x$- and $y$-axes in Fig. 1b). By unveiling this intricate self-assembly mechanism, we succeeded in obtaining supramolecular nanosheets with controlled areas and a distinct aspect ratio. Our original objective in designing **6FF** was to synthesize supramolecular nanosheets, similar to those of **6HH**, but with different aspect ratios[15], by changing the nature of the intermolecular interactions along the $y$-axis from van der Waals forces to a fluorophilic effect[16–21]. To our surprise, however, atomic-force microscopy (AFM) studies showed that **6FF** formed circular nanosheets with holes at their centers (Figs. 1c and 2c). Interestingly, each of these circular nanosheets consisted of a one-dimensional (1D) supramolecular polymer coiled into an Archimedean spiral (a spiral in which the consecutive turns are separated by a constant distance). Our initial analyses of the supramolecular polymerizations with mass-balance models failed to describe the aggregation process over a wide range of concentrations and implied that **6FF** might form different aggregates at low concentrations. Indeed, we discovered that **6FF** formed supramolecular concentric toroids under more-dilute conditions. Moreover, a mechanistic insight into the self-assembly process of 6FF permitted a controlled synthesis of the concentric toroids.

## Results

**Discovery of an unusual self-assembled structure.** When a hot solution of **6FF** in an aliphatic solvent was cooled to room temperature, a Soret band that was blue-shifted relative to that of the monomeric state was observed, suggesting the formation of an H-aggregate (i.e., face-to-face stacking) (Fig. 2a). Contrary to our expectations (see above), this result together with our previous studies indicated that **6FF** self-assembled to form 1D supramolecular polymers rather than 2D nanosheets[14,22–27]. When the solution was spin-coated onto a highly oriented pyrolytic graphite (HOPG) substrate and examined by AFM, we were surprised to find that the 1D supramolecular polymer coiled to form a series of circular nanosheets (Fig. 2c, d). Similar structures were observed

on other substrates, such as mica or silicon wafers, suggesting that the spirals were formed in solution without any influence of the substrate (Supplementary Fig. 7). As such, the self-assembly process of **6FF** is distinctly different from those of other supramolecular spirals reported in the literatures, which required the presence of 2D self-assembly fields, such as substrate surfaces or an air–water interface[28–32]. When we tested other solvents, we found that the spiral structures tended to form in linear alkanes, such as hexane or dodecane (Supplementary Fig. 8). No split Soret band was observed for **6FF** in any of the solvents examined, suggesting that it is incapable of forming short-slipping J-aggregates such as that of **6HH**, despite the structural similarity of these two monomers (Supplementary Fig. 9).

The spiral formed by **6FF** is tightly coiled with a constant separation distance between consecutive turns, a characteristic of so-called Archimedean spirals which can be described in polar coordinates as follows:

$$r = a + b\theta$$

In Fig. 2e, the center of the circular nanosheet was fixed at the origin of the polar coordinate and we defined $r$ as the distance between the origin and the spiral trajectory with the highest contrast in the AFM image. A plot of $r$ as a function of $\theta$ was fitted to values of $a$ and $b$ of 23.1 and 1.08 nm, respectively (Fig. 2f). Other structural features of the Archimedean spiral are summarized in Supplementary Fig. 10 and Supplementary Table 1. In brief, the thickness (4.3 nm) and circularity (0.96) were uniform, whereas the area was polydisperse [dispersity $Đ = 1.61$; $A_w = 38,500$ nm$^2$; $A_n = 24,000$ nm$^2$, where $Đ$ is the ratio of weight-average area ($A_w$) to the number-average area ($A_n$)].

Unlike previously reported supramolecular coils and helices[33,34], the curvature ($1/r$) of an Archimedean spiral is not a constant, indicating that the higher-order structure of **6FF** is not strongly dictated by the structure of the **6FF** molecule. It is reasonable to hypothesize that the 1D supramolecular polymer of **6FF** becomes tightly coiled so as to minimize the surface area of its fluorinated segments that is exposed to the aliphatic solvent[16–21]. This hypothesis is supported by our observation that the Archimedean spiral formed from **6FH**, which contains only one fluoroalkyl chain (Fig. 1a), was less stable than that formed from **6FF** (discussed later). If tight packing of the side chains drives the formation of the spiral, the fluoroalkyl chains should be oriented in-plane of the circular nanosheets, which in turn suggests that the constant separation distance between consecutive turns in the spiral should match the length between the two fluorinated side chains in **6FF**. However, the separation between the turns ($2\pi b = 6.8$ nm) was about double the distance between the side chains (3.5 nm; Supplementary Fig. 11). Thus, the surface visualized by AFM is not satisfactorily explained by the proposed molecular packing geometry of **6FF** in the Archimedean spiral.

To address this discrepancy between the spacing and the side-chain distance, we examined the Archimedean spiral by scanning transmission electron microscopy (STEM) and by small-angle X-ray scattering (SAXS). Interestingly, the separation distance between consecutive turns was determined to be 3.1 nm by annular dark field (ADF)–STEM (Fig. 2g and Supplementary Fig. 12). In addition, SAXS measurements on a solution sample showed a similar periodicity of 3.3 nm (Supplementary Fig. 13). Much to our surprise, the evidence as a whole suggests that our Archimedean spiral is double-stranded (Fig. 2h); namely, the spiral is a layered structure in which another Archimedean spiral is present on the bottom of the circular nanosheet. In fact, when we inspected defect and edge domains by AFM, a thin layer was confirmed beneath the spiral (Supplementary Fig. 14). Thus, adjacent 1D supramolecular polymers in the spiral are displaced alternately upwards and downwards with respect to the substrate normal, permitting the

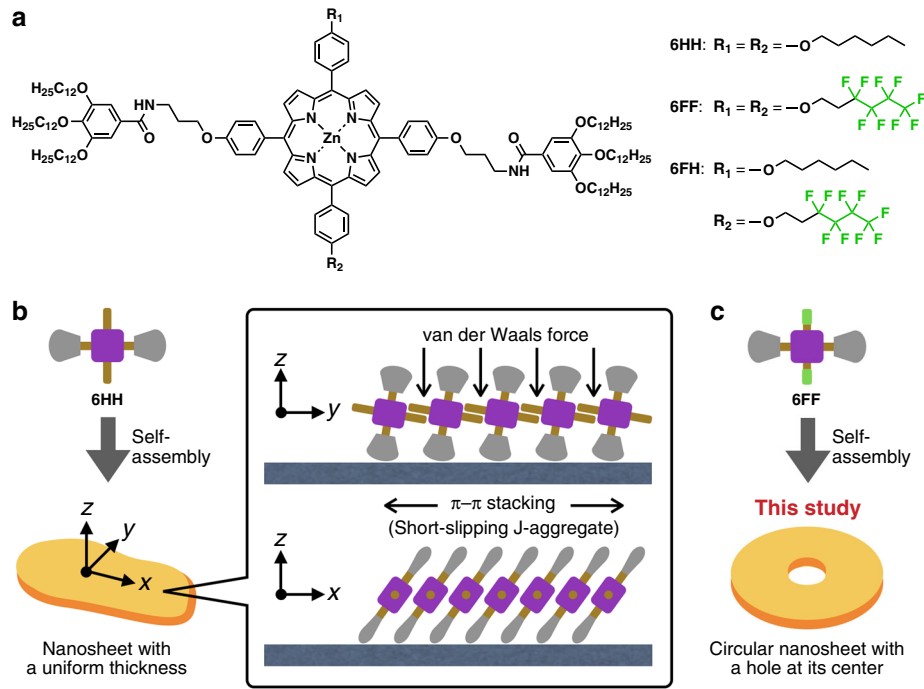

**Fig. 1 Molecular design and self-assembly. a** Molecular structures of **6HH**, **6FF**, and **6FH** used in this study. **b** Schematic representation of self-assembly of **6HH** to form a supramolecular nanosheet. Before our present study, we expected that **6FF** would form a supramolecular nanosheet, similar to that formed by **6HH**, but with a different aspect ratio as a result of differences in intermolecular interactions along the *y*-axis. **c** Schematic representation of a supramolecular circular nanosheet with a hole at its center formed from **6FF**.

fluorinated alkyl chains to overlap each other. In such a packing model, the thickness of the nanosheet (4.3 nm) appears to be inconsistent with the molecular length (6.9 nm; Supplementary Figs. 11 and 15). We infer that the dodecyl chains at the periphery of **6FF** adopt a folded gauche conformation and fill the space between the spirals; this was corroborated by Fourier-transform infrared spectral measurements (Supplementary Figs. 15 and 16).

**Self-assembly mechanism of archimedean spiral**. Cooling of a hot solution of **6FF** in dodecane resulted in the formation of double-stranded Archimedean spirals irrespective of the cooling rate (Supplementary Fig. 17). Temperature-dependent absorption spectral measurements showed the presence of three distinct spectral signatures (Fig. 3a). More precisely, on cooling **6FF** from its molecularly dissolved state ($\lambda_{max}$ = 420 nm), a red-shifted J-aggregate Soret band appeared at 434 nm as a shoulder (373 → 323 K), which was subsequently replaced by the blue-shifted H-aggregate Soret band at 403 nm (323 → 298 K). Previously, we characterized this J-aggregates as an off-pathway intermediate to the thermodynamically stable H-aggregates[22,23]. When a hot solution of **6FF** was rapidly quenched in an ice bath (30 s), the J-aggregate state could be kinetically retained as a metastable state; this subsequently underwent transformation into an H-aggregated Archimedean spiral (Fig. 3b). Time-dependent changes in the absorbance exhibited sigmoidal kinetics (Fig. 3c). AFM measurements in the early stages of the self-assembly captured the presence of smaller spirals with several turns (Fig. 3d, e), which then grew larger at the expense of the J-aggregates (Fig. 3f, g). These results indicate that the Archimedean spirals are formed through a cooperative nucleation–growth process. The area of the Archimedean spirals obtained after time-dependent evolution was polydisperse (Đ = 1.40, Supplementary Fig. 18), suggesting that the primary nucleation process was occurring concurrently with the growth process, and its propagation was not controlled kinetically.

Recently, mass-balance models have proven to be powerful tools for elucidating supramolecular polymerization mechanisms[35–38]. To quantify the thermodynamic parameters of the formation of the Archimedean spiral, a mass-balance model incorporating competition between the isodesmic J-aggregation and the nucleated H-aggregation (see Supplementary Information for full details) was fitted to the temperature-dependent changes in absorbance[35]. To probe the formation of both types of aggregate, the absorbances at 434 and 403 nm were monitored (Fig. 4a). Molar absorption coefficients of **6FF** in monomeric, J-aggregate, and H-aggregate forms were determined by separate experiments (Supplementary Fig. 19). The cooling curves were obtained for six different concentrations at a sufficiently slow cooling rate (−0.5 K/min). As the Archimedean spiral was prone to precipitate at lower temperatures, as confirmed by a Tyndall effect (Supplementary Fig. 20), we only used data points for temperatures at which <65% of **6FF** molecules aggregated to form H-aggregates (i.e., the filled marks in Fig. 4a). However, a global fit that treated the data sets of six different concentrations simultaneously with a single set of thermodynamic parameters proved to be unsuccessful (Supplementary Fig. 22). Faced with this problem, we realized that changes in absorbance measured at higher (reddish plots) or lower concentrations (bluish plots) showed slightly different trends (Fig. 4a), suggesting the presence of two very similar, yet subtly different aggregation pathways (*vide infra*). As the very similar spectral properties of these aggregates prevented us to successfully fit the data, we selected appropriate data sets where only a single aggregate is observed. To our delight, global fits using data sets obtained at higher concentrations (10, 12.5, and 15 μM) gave accurate thermodynamic parameters for the formation of the Archimedean spiral (Fig. 4b and Supplementary Fig. 23). Porphyrin **6FH** was also capable of forming Archimedean spiral in dodecane, a process that was analyzed accordingly (Supplementary Figs. 24–26). Table 1 summarizes the resulting values.

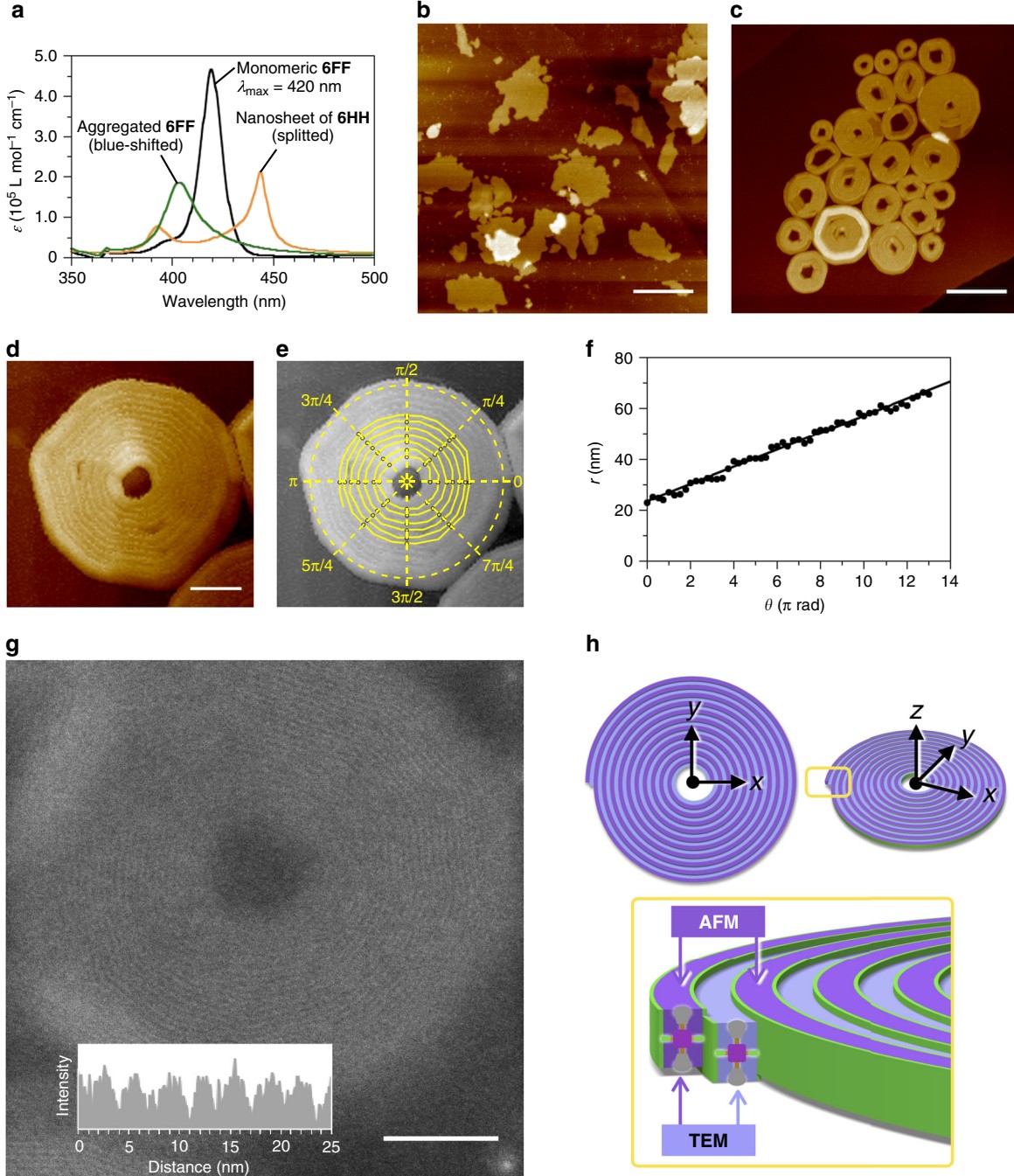

**Fig. 2 Self-assembly of 6FF. a** Absorption spectra of supramolecular nanosheets of **6HH** (50 μM in methylcyclohexane at 283 K), Archimedean spirals of **6FF** (12.5 μM in dodecane at 298 K), and monomeric **6FF** (12.5 μM in dodecane at 373 K). Spectra of the aggregates were measured under conditions of concentration and temperature where the degree of aggregation is close to unity. **b** AFM image of supramolecular nanosheets of **6HH**. Scale bar = 200 nm. **c–e** AFM images of Archimedean spirals of **6FF**. Scale bar = 200 nm (**c**) and 50 nm (**d**). The images displayed in **d** and **e** are inverted so that the spiral is shown as anticlockwise in the polar coordinate. **f** Plot of $r$ against $\theta$ for the points in the polar coordinates shown in **e**. **g** ADF–STEM image of the Archimedean spiral of **6FF**. Scale bar = 50 nm. Inset: cross-sectional histogram. **h** Schematic representation of self-assembly of **6FF** to form a supramolecular double-stranded Archimedean spiral.

The thermodynamic parameters obtained from the fits show that both the J-aggregates and the Archimedean spirals of **6FF** are characterized by a stronger gain in enthalpy compared with aggregates of **6FH**. For the off-pathway isodesmic J-aggregates, the difference in thermodynamic parameters is minor, suggesting that the aggregation in these off-pathway aggregates is primarily driven by interactions between the porphyrins. In contrast, the thermodynamic parameters for the formation of the H-aggregated Archimedean spiral are markedly different. As a result, the thermal stability of the Archimedean spirals of **6FF** is slightly higher than that of **6FH**, as indicated by the elongation temperatures of the spirals in 12.5 μM solutions of **6FF** and **6FH** at 329 and 325 K, respectively (Supplementary Fig. 28), and by their critical aggregation concentrations of $1.7 \times 10^{-8}$ and

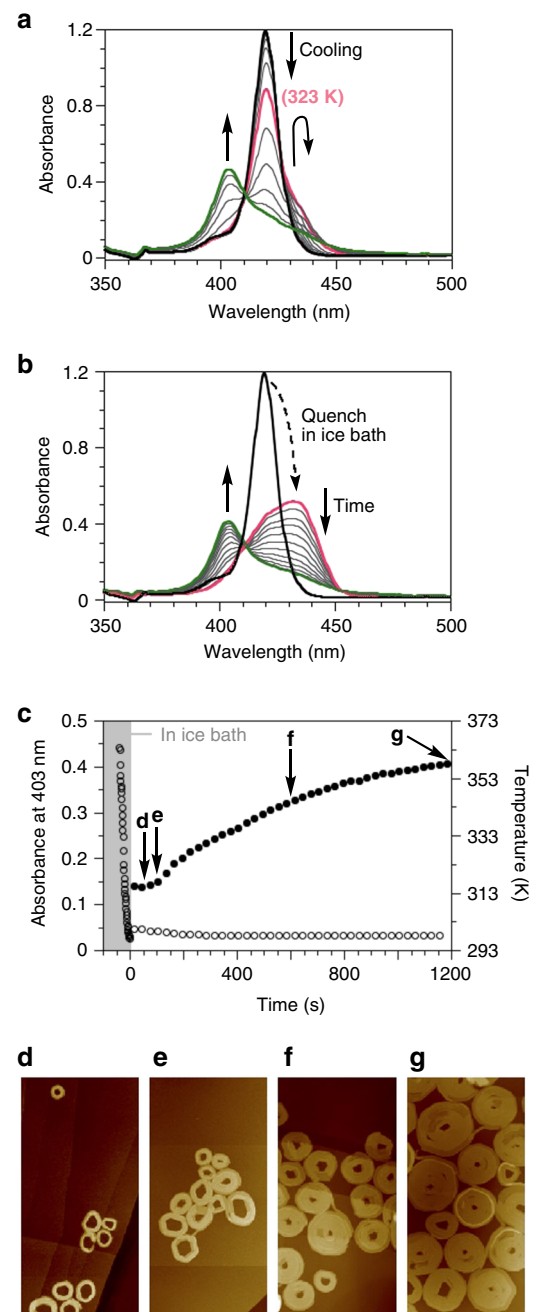

**Fig. 3 Molecular self-assembly of 6FF. a** Temperature-dependent changes in the absorption spectrum of a solution of **6FF** in dodecane: [**6FF**] = 12.5 μM, 373 → 298 K, −5.0 K/min. **b** Time-dependent changes in the spectrum of a solution of **6FF** prepared by quenching a hot dodecane solution (373 K, shown in black line) in an ice bath for 30 s: [**6FF**] = 12.5 μM, measured at 298 K. **c** Plots of the absorbance at 403 nm (filled circle) and the temperature of the solution (open circle) over time measured at 298 K. **d–g** AFM images of the self-assembled **6FF** observed during its propagation. Scale bar = 200 nm. The AFM specimens were prepared from solution at the times indicated by the arrows in **c**: at **d** 50 s, **e** 100 s, **f** 600 s, and **g** 1200 s.

$9.5 \times 10^{-8}$ M at 298 K, respectively (based on Supplementary Eq. S2). An increased thermal stability of supramolecular aggregates in the presence of an increased amount of fluorinated side chains has also been observed in liquid crystals[18]. In addition

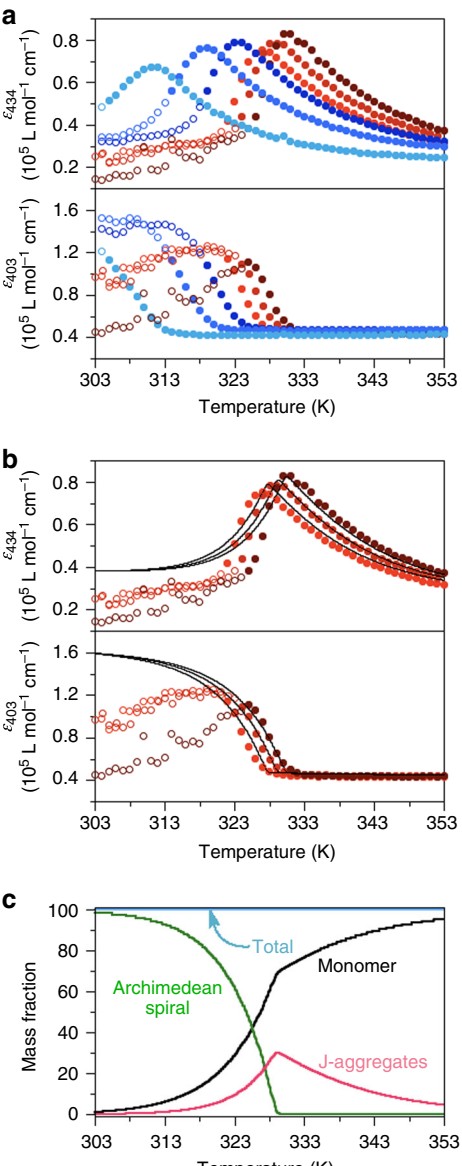

**Fig. 4 Thermodynamic analysis. a** Plots of molar absorption coefficients at 434 and 403 nm as a function of temperature: [**6FF**] = 2.5, 5.0, 7.5, 10.0, 12.5, 15.0 μM; −0.5 K/min. It is noteworthy that the plots obtained at lower concentrations (bluish marks) and at higher concentrations (reddish marks) showed slightly different trends. **b** The two-pathway thermodynamic model was fitted (solid lines) to the plots indicated by the reddish filled marks in **a** (i.e., the data points obtained at higher concentrations). **c** Calculated temperature-dependent speciation plot of a 12.5 μM solution of **6FF**. The speciation plot shows the distribution of monomers over the free monomeric, J-aggregated, and H-aggregated (Archimedean spiral) states (see also Supplementary Fig. 27).

to the stronger enthalpic component for the formation of aggregates of **6FF**, the entropic penalty for the formation of aggregates of **6FF** is also considerably more negative than that for the aggregation of **6FH**. This increased entropic penalty suggests that the side chains in the Archimedean spirals of **6FF** are more organized and structured compared with those of **6FH**. Presumably, the poorer solubility of the fluorinated side chains in aliphatic environments drives this stronger organization. Thus, by balancing the degree of fluorination of the side chains, fine control of thermal properties of the supramolecular structures is attainable.

**Table 1 Thermodynamic parameters.**

|  | 6FF | 6FH |
|---|---|---|
| J-aggregate, monitored at 434 nm |  |  |
| $\Delta H_{iso}$ (kJ mol$^{-1}$) | −95.8 | −81.0 |
| $\Delta S_{iso}$ (J mol$^{-1}$ K$^{-1}$) | −196.1 | −156.2 |
| $\Delta G_{iso}$ (kJ mol$^{-1}$) at 308 K | −35.4 | −32.9 |
| H-aggregated Archimedean spiral, monitored at 403 nm |  |  |
| $\Delta H_e$ (kJ mol$^{-1}$) | −125.3 | −93.7 |
| $\Delta S_e$ (J mol$^{-1}$ K$^{-1}$) | −271.1 | −180.2 |
| $\Delta G_e$ (kJ mol$^{-1}$) at 308 K | −41.8 | −38.2 |
| $\sigma^a$ at 308 K | $6.6 \times 10^{-6}$ | $1.7 \times 10^{-4}$ |

The values obtained from a fit of the mass-balance model of the polymerization of **6FF** and **6FH** to form off-pathway J-aggregates and H-aggregated Archimedean spirals.
$^a$The cooperativity parameter $\sigma$ is calculated from the nucleation penalty NP. See Supplementary Information for details.

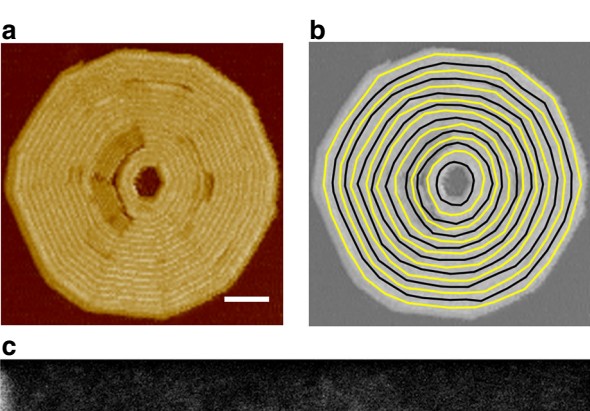

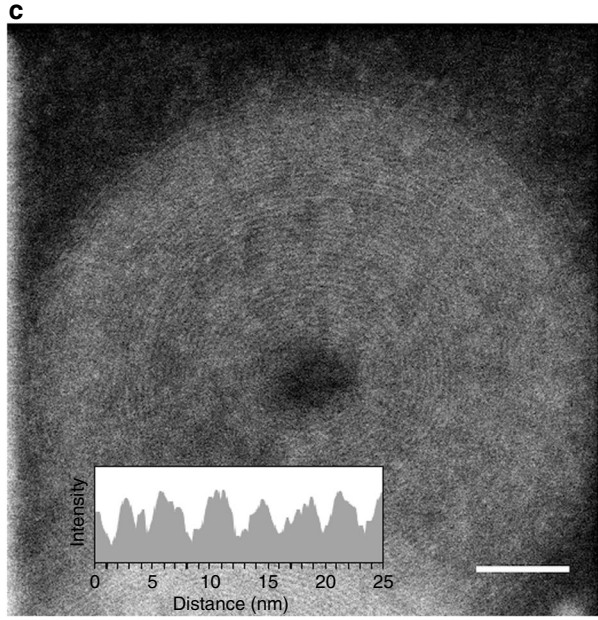

**Fig. 5 Concentric toroids of 6FF. a, b** AFM images and **c** ADF–STEM image of concentric toroids obtained by cooling a hot solution of **6FF** (7.5 μM) in dodecane. Scale bar = 50 nm (**a, c**).

**Thermodynamic analysis implies another structure.** The 'unsuccessful' global fit (see above) was indicative of the possible presence of a different self-assembled structure at lower concentrations (<10 μM), which we investigated by using AFM. As shown in Fig. 5, we observed the presence of circular nanosheets with holes at their centers, closely resembling the Archimedean spiral described above. However, the 1D supramolecular polymers in each turn were found to be closed and nested, resulting in

the formation of a concentric toroid (Fig. 5a, b). Again, the separation distance between the toroids determined by AFM was larger than the length between the two fluorinated side chains in **6FF** (3.5 nm), but ADF–STEM visualized the true separation distance between the consecutive toroids to be 3.2 nm (Fig. 5c and Supplementary Fig. 29). Absorption spectral measurements revealed that the concentric toroids consisted of H-aggregates, as in the case of the Archimedean spiral (Supplementary Fig. 30). However, a global fit to the two-pathway model by using only data points obtained at lower concentrations (2.5, 5.0, and 7.5 μM) did not yield a unique solution (Supplementary Fig. 31). These results indicate that the Archimedean spirals and the concentric toroids are similar in terms of their molecular packing, but differ in terms of their self-assembly mechanisms. As such, these two structures are in a unique relationship from the viewpoint of supramolecular polymorphism[14,39–42]. Under a diluted condition (5.0 μM), 47 out of 50 circular nanosheets were concentric toroids (94%), while under a concentrated condition (12.5 μM), 46 of them were double-stranded Archimedean spirals (92%). This result demonstrates that difference in the self-assembly mechanism achieves remarkable selectivity even in the creation of such similar self-assembled structures.

**Controlled synthesis of concentric toroids.** We reasoned that dilute conditions led to the preferential formation of toroids rather than spirals. The fluorinated surfaces of the resulting primary toroids then serve as templates for the successive formation of new toroids through secondary nucleation (Fig. 6a). This notion was corroborated by the fact that under the same conditions, **6FH** failed to form multiple concentric toroids due to its weaker fluorophilic effect (Supplementary Fig. 32).

We rarely observed unclosed toroids, indicating that 1D elongation along the circumference is much faster than secondary nucleation (Fig. 6a, box). Nevertheless, because of the unique topology of the toroids, 1D elongation terminates at every toroid closure. As a result, these two propagation steps should occur alternately. We envisioned that this uncommon 2D growth mechanism permits the controlled synthesis of concentric toroids, provided that the number of the primary toroids can be defined.

To shift the primary nucleation process to the kinetic regime, we applied a solvent-mixing protocol using a good and a poor solvent[6,43]. In a cuvette, dodecane (a poor solvent) was gently added over a solution of monomeric **6FF** in toluene (a good solvent) and the mixture was then shaken vigorously for a second and left at 298 K (see the "Methods" section). AFM measurements confirmed that supramolecular concentric toroids were reproducibly obtained by this protocol. In addition, the presence of the good solvent apparently prevented precipitation of the concentric toroids.

Absorption spectral measurements indicated that the J-aggregates were instantaneously formed upon solvent mixing and then transformed over time to form the H-aggregated concentric toroids (Fig. 6b, c). To capture the self-assembly process, specimens for AFM observation were prepared at various times, as indicated by the arrows in Fig. 6c. Interestingly, the area ($A_n$) of the concentric toroids increased proportionally against the degree of **6FF** in concentric toroids determined by absorption spectral changes (Fig. 6d–h, j, k). This result indicates that the number of the concentric toroids is practically constant throughout the propagation process, which is analogous to a controlled "chain-growth polymerization" process. Consequently, the solvent-mixing protocol afforded concentric toroids with a narrower polydispersity ($Đ = 1.18$) than that of toroids obtained by the temperature-change protocol ($Đ = 1.44$; Fig. 5a and Supplementary Fig. 33). More importantly, addition of

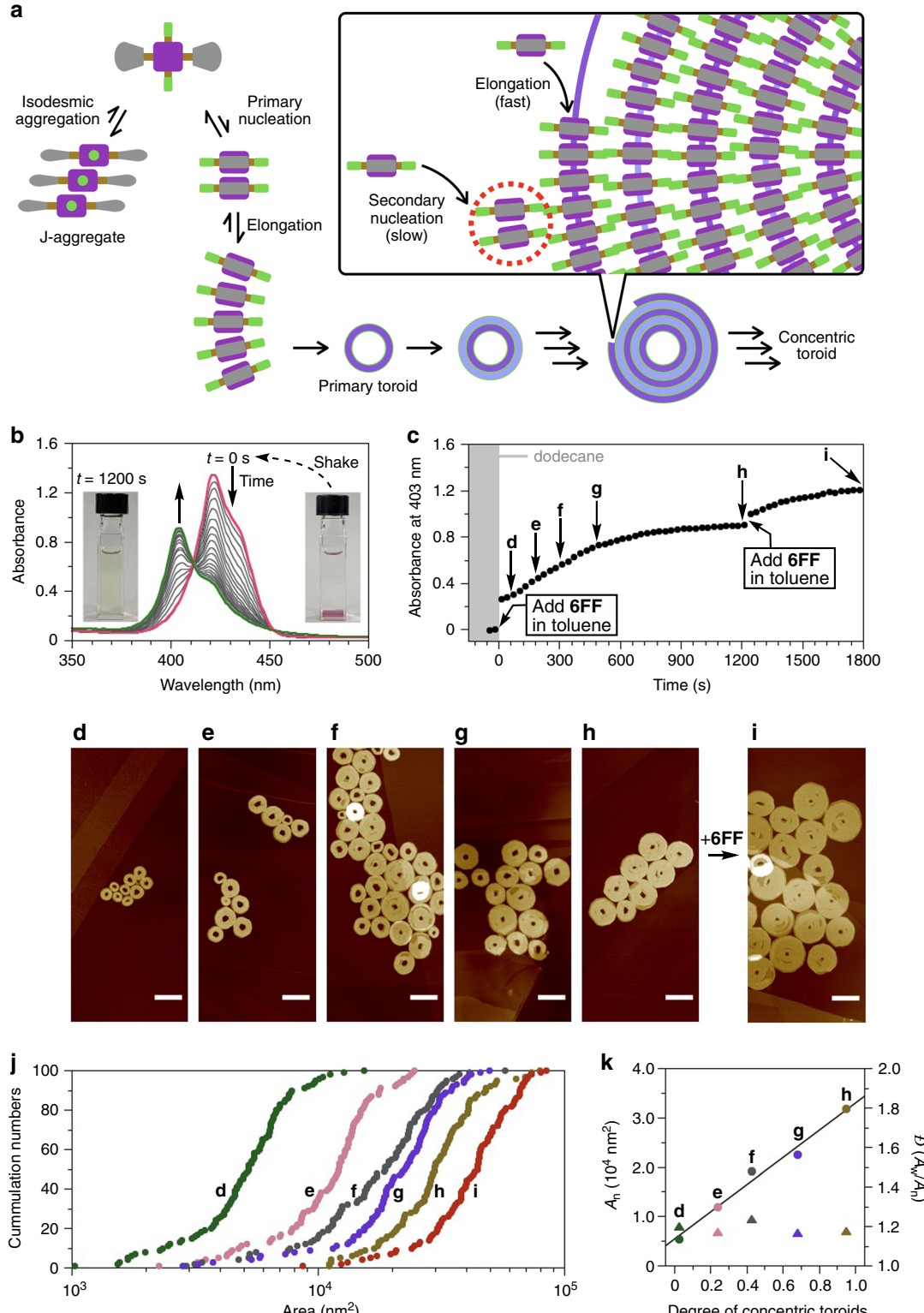

**Fig. 6 Controlled synthesis of concentric toroids. a** Schematic illustration of the formation of the concentric toroids. It is noteworthy that new toroids are formed not only at the outer circumference but also at the inner circumference of the primary toroid, as suggested by the decrease in size of the hole observed in the AFM images. **b** Time-dependent absorption spectral changes observed on mixing a toluene solution of **6FF** with dodecane. [**6FF**] = 5 μM, toluene–dodecane = 1:19 (v/v), 298 K. **c** Plot of the absorbances at 403 nm as functions of time. **d–i** AFM images of concentric toroids obtained during the time-dependent evolution at **d** 60 s, **e** 180 s, **f** 300 s, **g** 480 s, **h** 1200 s, and **i** 1800 s (i.e., after the incremental addition of a **6FF** solution). Scale bar = 200 nm. **j** Cumulative histograms of the area of concentric toroids obtained after the first **d–h** and the second addition **i** of a toluene solution of **6FF** to dodecane. The data were collected by tracing 100 objects in AFM images. **k** Plots of $A_n$ (circle) and Đ ($A_w/A_n$, triangle) of the concentric toroids as a function of the degree of the concentric toroids **d–h**.

monomeric **6FF** in toluene to the resultant solution of concentric toroids further increased $A_n$ and $A_w$ to 44,000 and 49,700 $nm^2$, respectively, while maintaining the low $Đ$ value (1.13) (Fig. 6i, j). Thus, mechanistic insights into the self-assembly permitted the controlled synthesis of an unusual supramolecular structure.

Previously, George and colleagues[44,45] have successfully achieved 1D supramolecular polymers with a controlled length by similar seed formation in situ (i.e., without adding seeds externally). However, in practice, it would be difficult to extend this approach to 2D self-assembly that consists of 1D supramolecular polymerization and lateral growth, because each propagation process occurs with significantly different kinetics[46]. We thus assert that the uncommon 2D self-assembly mechanism that stems from the unique concentric toroid structure played an important role in controlling the area of the circular nanosheets (Fig. 6a). Recently, the topologies of self-assembled structures have been shown to influence their functions and properties; e.g., Yagai and colleagues[47] demonstrated an effect of topology on the kinetic stability of a supramolecular polymer. Our study illustrates a new topological effect that might be exploited in precise noncovalent syntheses[11].

## Discussion

This study illustrates the richness of self-assembled structures obtainable at higher hierarchical levels. It is remarkable that new and unusual self-assembled structures, such as double-stranded Archimedean spiral, could be obtained from a simple porphyrin molecule despite our previous systematic investigations on similar porphyrin derivatives[14,15,22–27]. By using mass-balance models, we found that the higher-order structure was stabilized by a lateral fluorophilic effect; this was consistent with the molecular packing proposed on the basis of AFM, ADF–STEM, and SAXS results. Intriguingly, however, a global fit of a data set including data obtained at low concentrations failed; this result led us to discover another unusual structure, that of supramolecular concentric toroids. Having established the topologically defined self-assembly mechanism, this study also demonstrates a new topological effect that could pave the way to controlled noncovalent syntheses. So far, complex structures have been created from inorganic materials[48–50] and block polymers[51–53], and similarly, many unknown opportunities in molecular self-assembly can be expected at larger scales, reflecting the recent advance in the understanding of the self-assembly mechanisms. Finally, it is interesting to note that a similar peculiar structure is used to achieve specific functions in some devices[54] as well as in a living system[55].

## Methods

**Scanning transmission electron microscopy**. Drops of the solutions were dispersed on copper grids covered with 6 nm-thick amorphous carbon films. The grids were dried at room temperature. ADF–STEM observations were performed using electron microscopes (Titan Cubed and Themis Z; Thermo Fisher Scientific, Inc.) operated at 300 kV. The probe current and convergence semi-angle were set as 5–8 pA and 18–25 mrad. The collection semi-angle range were set as 50–200 (Fig. 2f and Supplementary Fig. 12) and 30–190 mrad (Fig. 5c and Supplementary Fig. 29).

**Controlled synthesis of concentric toroids**. In a 10 mm × 10 mm cuvette, a solution of **6FF** (100 μM, 150 μL) in toluene was placed. On this solution, dodecane (2850 μL) was gently added, resulting in two separated organic layers. The cuvette was shaken vigorously for a second and then transferred to the sample compartment of a ultraviolet-visible (UV-Vis) spectrophotometer, which was kept at 298 K: the concentration and solvent ratio were [**6FF**] = 5 μM and toluene–dodecane = 1:19 (v/v), respectively. After 20 min, a solution of **6FF** (50 μM, 170 μL) in toluene was added to the resultant solution and UV-Vis spectral changes were continuously measured: the concentration and solvent ratio were [**6FF**] = 7.4 μM and toluene–dodecane = 1:9 (v/v), respectively. Although the absorption spectral change was monitored at given times (Fig. 6c), a solution (5 μL) was sampled and spin-coated on a HOPG substrate, which was dried under vacuum for AFM measurements.

## Data availability

The authors declare that the data supporting the findings of this study are available within the paper and its Supplementary Information file. All other information is available from the corresponding authors upon reasonable request.

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

## Acknowledgements

This research was supported by KAKENHI (JP15H05483 and JP19K05592 for K.S.), Scientific Research on Innovative Areas "Soft crystals: science and photofunctions of flexible response systems with high order" (H2004682 for K.S.), "π-System figuration: control of electron and structural dynamism for innovative functions" (JP26102009 for M.T.), and the Nanotechnology Network Project from the Ministry of Education, Culture, Sports, Science and Technology, Japan. The SAXS measurements were performed at the Photon Factory of High Energy Accelerator Research Organization (Approval numbers 2018P010). M.F.J.M. and E.W.M. acknowledge financial support from NWO (TOP-PUNT Grant 10018944) and the Dutch Ministry of Education, Culture, and Science (Gravitation program 024.001.035).

## Author contributions

K.S. designed the research. N. Sasaki synthesized the molecules and investigated their self-assembly. M.F.J.M. analyzed self-assembly based on the mass-balance models. J.K. performed the STEM experiments. The molecular packings were analyzed by N. Shioya, T.S., and T.H. using FT-IR and by H.T., R.H., N. Shimizu, and S.-i.A. using SAXS. N. Sasaki, M.F.J.M., T.F., E.W.M., M.T., and K.S. discussed the self-assembly mechanism. N. Sasaki, M.F.J.M., and K.S. wrote the manuscript with input from all authors.

## Competing interests

The authors declare no competing interests.
