## [Peer Review File · Nature Communications]

REVIEWER COMMENTS

Reviewer #1 (Remarks to the Author):

The manuscript by Meijer, Takeuchi, Sugiyasu and coworkers presents the formation of hierarchically self-assembled structures at a high level from porphyrin derivatives. The unexpected self-assembled morphologies, such as one-dimensional (1D) supramolecular polymers coiled into Archimedean spirals and supramolecular concentric toroids formed at low concentrations, are rare and very interesting. The synthesis, self-assembly, analysis of the assembly mechanisms, and the thermodynamic analysis are well described in the manuscript. Although the original purpose of the author's design of 6FF was to synthesize supramolecular nanosheets similar to 6HH, the results are unexpected and the studies are very clear and beautiful. As the presented results would be great interesting to the broad readership of Nature Communications, I recommend the publication of this manuscript after minor revision. There are a few comments and suggestions, which I would like the authors to consider upon revising their manuscript:

1. The critical aggregation concentration is an important parameter to an aggregation self-assembly system. The CACs of 6FF and 6FH should be measured, which may provide a deeper understanding of concentration effects on the aggregation processes.
2. As the zinc complexes of porphyrin monomer derivatives were used for the self-assembly, is it necessary to specifically complex zinc ion on the porphyrin monomers? Does the presence of zinc ion have a specific role for the supramolecular assembly?
3. The stability of the self-assembly is as important as its structure, which may affect the performance and applications of the materials. How about the stability of the formed Archimedean spirals and concentric toroids. How long can the self-assemblies be stable after they are formed?

Reviewer #2 (Remarks to the Author):

In the present work, Sasaki et al. describe the self-assembly of a porphyrinato zinc derivative bearing fluorinated side chains. They observe two supramolecular polymorphs, namely Archimedean spirals at high concentrations and concentric toroids at low concentrations. The unprecedented and rather unusual nanostructures were characterized by atomic force microscopy (AFM), annular dark field scanning transmission electron microscopy (ADF-STEM) as well as small-angle X-Ray scattering. Furthermore, UV/vis spectroscopic studies in combination with mass-balance models provided mechanistic insight into the thermodynamic and kinetic characteristics of Archimedean spiral and toroid formation. These in-depth studies allowed the authors to find conditions for the exclusive synthesis of the second supramolecular polymorph, i.e. concentric toroids, with small dispersities.

This is a very impressive contribution because it does not only show beautiful and unusual nanostructures but provides excellent insights into their structural evolution. Thus, the presented work provides a beautiful example of hierarchical self-assembly and deepens the knowledge on non-covalent supramolecular synthesis. Publication after minor changes is recommended.

Minor comments:

Main Manuscript

- Page 5: In the molecular structure of the zinc porphyrin derivative, on the left side the connectivity of the dodecyloxy chains should be altered to "H25C12O" instead of "C12H25O" to make the connectivity clearer. This should be also corrected in the reaction Schemes in the SI.
- Regarding the general equation of the Archimedean spiral: Please give a short description what the variables mean.
- Page 12: Upon (rapid) cooling, kinetically metastable, off-pathway J-aggregates are formed by 6FF. Is the cooling rate of 0.5 K/min slow enough to ensure that the self-assembly process is under equilibrium conditions? If not, this might also explain the small deviations of the fit

(thermodynamic mass-balance model) from the experimental data presented in Figure 4b.

- Figure 4: On page 27 in the SI it is stated that the temperature-dependent UV/vis spectra have been calculated. A comparison of at least one experimentally obtained UV/vis spectrum at a certain temperature and the calculated UV/vis trace within the same graph would be nice to assess the quality of the fit.

- Figure 4a: Please explain in more detail why the apparent extinction coefficients at lower concentrations cannot be fitted with the mass balance model although the absorption spectra of the spirals and toroids are almost identical and the molecular arrangement within the spirals and toroids is similar. Is this a result of different secondary nucleation mechanisms of the spirals and toroids or is it more likely that the toroids are formed by a different aggregation mechanism?

- Page 14: Please provide for 6HF also a plot of the temperature-dependent changes of the extinction coefficients at a certain wavelength as done in Figure 4a and b in the SI. Otherwise the elongation temperature of a 12.5 μM solution of 6HF of $T = 325 \text{ K}$ cannot be assessed.

- Please comment: Is the toroid structure also a double layer structure as the spiral?

- Figure 6a: This cartoon suggests that the primary toroid is the most inner one, i.e. that growth takes place only at the outside. However, my impression is that also filling on the inner side takes place (see Figure 6e to 6g). Please explain.

Supporting Information

- Integrals should be added in the NMR spectra

- In the reaction scheme, the dodecyloxy chain should be altered to "H25C12O" instead of "C12H25O" to make the connectivity clearer.

Reviewer #3 (Remarks to the Author):

Sugiyasu and Takeuchi use a Zn-porphyrin with fluorinated and non-fluorinated side-chains to investigate unique double stranded Archimedean spirals and concentric toroids. The level of control is extremely impressive, and the complexity of the assembly process that arises from these partially known molecular designs is unpredictable. This is especially true for the concentration dependent morphology and assembly mechanisms. The paper will appeal to the supramolecular, organic materials and soft matter communities and thus have a big impact. The AFM microscopy, in combination with SAXS analysis are impeccable. The manuscript is further supported by a complementary modelling studies from the Eindhoven lab, which is essential to undermine the mechanistic aspect. The strong dependency on the solvent, further supports the critical role of 'fluorophilic' effect. I support publication as is.

Reviewer #1 (Remarks to the Author):

The manuscript by Meijer, Takeuchi, Sugiyasu and coworkers presents the formation of hierarchically self-assembled structures at a high level from porphyrin derivatives. The unexpected self-assembled morphologies, such as one-dimensional (1D) supramolecular polymers coiled into Archimedean spirals and supramolecular concentric toroids formed at low concentrations, are rare and very interesting. The synthesis, self-assembly, analysis of the assembly mechanisms, and the thermodynamic analysis are well described in the manuscript. Although the original purpose of the author's design of 6FF was to synthesize supramolecular nanosheets similar to 6HH, the results are unexpected and the studies are very clear and beautiful. As the presented results would be great interesting to the broad readership of Nature Communications, I recommend the publication of this manuscript after minor revision. There are a few comments and suggestions, which I would like the authors to consider upon revising their manuscript:

Reply Thank you very much for your highly positive comments. We have revised the manuscript according to your comments.

1. The critical aggregation concentration is an important parameter to an aggregation self-assembly system. The CACs of 6FF and 6FH should be measured, which may provide a deeper understanding of concentration effects on the aggregation processes.

Reply We agree that the critical aggregation concentration is an important characteristic of self-assembling systems. The elongation temperature of the supramolecular polymers gives the information required to assess the concentration effects. At the elongation temperature, the free monomer concentration is equal to the inverse of the elongation constant of those polymers. Since we could successfully obtain the thermodynamic parameters for the Archimedean spirals that are dominant at high concentration, the critical aggregation concentration at any temperature can in principle be calculated. We adapted the manuscript to make this connection clearer. This was added in the revised manuscript.

(page 14, line 7) As a result, the thermal stability of the Archimedean spirals of **6FF** is slightly higher than that of **6FH**, as indicated by the elongation temperatures of the spirals in 12.5 μM solutions of **6FF** and **6FH** at 329 and 325 K, respectively (Supplementary Fig. 28), and by their critical aggregation concentrations of 1.7×10^{-8} and 9.5×10^{-8} M at 298 K, respectively.

2. As the zinc complexes of porphyrin monomer derivatives were used for the self-assembly, is it necessary to specifically complex zinc ion on the porphyrin monomers? Does the presence of zinc ion have a specific role for the supramolecular assembly?

Reply We found that a copper analogue can also form these nanostructures. It should be interesting to extend this research using these different metal porphyrins. However, we consider the effect of metal ions is beyond the scope of this study, and we will publish such results in due course.

3. The stability of the self-assembly is as important as its structure, which may affect the performance and applications of the materials. How about the stability of the formed Archimedean spirals and concentric toroids. How long can the self-assemblies be stable after they are formed?

Reply We have kept the sample of the concentric toroids prepared on September 18, 2018 under vacuum. We could still observe the circular nanosheets (**Figure R1**). However, the stability in this respect cannot be quantitatively assessed. Obviously, many factors such as light, temperature, oxygen would affect. Because we discussed the thermodynamic stability in detail, we would like to refrain from mentioning the long-term stability in this manuscript.

Figure R1. AFM image of concentric toroids prepared 20 months ago, measured on May 26, 2020.

Reviewer #2 (Remarks to the Author):

In the present work, Sasaki et al. describe the self-assembly of a porphyrinato zinc derivative bearing fluorinated side chains. They observe two supramolecular polymorphs, namely Archimedean spirals at high concentrations and concentric toroids at low concentrations. The unprecedented and rather unusual nanostructures were characterized by atomic force microscopy (AFM), annular dark field scanning transmission electron microscopy (ADF-STEM) as well as small-angle X-Ray scattering. Furthermore, UV/vis spectroscopic studies in combination with mass-balance models provided mechanistic insight into the thermodynamic and kinetic characteristics of Archimedean spiral and toroid formation. These in-depth studies allowed the authors to find conditions for the exclusive synthesis of the second supramolecular polymorph, i.e. concentric toroids, with small dispersities.

This is a very impressive contribution because it does not only show beautiful and unusual nanostructures but provides excellent insights into their structural evolution. Thus, the presented work provides a beautiful example of hierarchical self-assembly and deepens the knowledge on non-covalent supramolecular synthesis. Publication after minor changes is recommended.

Reply Thank you very much for your highly positive comments. We have revised the manuscript according to your comments.

Minor comments:

Main Manuscript

- Page 5: In the molecular structure of the zinc porphyrin derivative, on the left side the connectivity of the dodecyloxy chains should be altered to "H25C12O" instead of "C12H25O" to make the connectivity clearer. This should be also corrected in the reaction Schemes in the SI.

Reply We have corrected the structures.

- Regarding the general equation of the Archimedean spiral: Please give a short description what the variables mean.

Reply We have added a short description regarding the equation of the Archimedean spiral:

(page 6, line 19) In Fig. 2e, the center of the circular nanosheet was fixed at the origin of the polar coordinate, and we defined r as the distance between the origin and the spiral trajectory with the highest contrast in the AFM image.

- Page 12: Upon (rapid) cooling, kinetically metastable, off-pathway J-aggregates are formed by 6FF. Is the cooling rate of 0.5 K/min slow enough to ensure that the self-assembly process is under equilibrium conditions? If not, this might also explain the small deviations of the fit (thermodynamic mass-balance model) from the experimental data presented in Figure 4b.

Reply The reviewer is right to point out that a small amount of kinetic control would introduce small deviations of the fit. If kinetic control in the self-assembly at a cooling rate of 0.5 K/min would be dominant, a global fit of the thermodynamic model to a dataset containing multiple concentrations would not work. However, the fit gives good results for all concentrations with a single set of thermodynamic parameters. We infer from the good agreement between the data and model, using a single set of parameters, that at these temperatures and a cooling rate of 0.5 K/min, the supramolecular polymerization is in thermodynamic control, or very close to thermodynamic control (**Figure R2**). Experiments to probe the polymerization at much lower cooling rates were unsuccessful due to precipitation of the aggregates, which prevents us from looking to the dynamic equilibria between the dissolved aggregates selectively.

Figure R2. Plot of absorbance at 403 nm as a function of temperature, obtained at different cooling rates. We can safely assume that the supramolecular polymerization is in thermodynamic control, or very close to thermodynamic control with a cooling rate of 0.5 K/min.

- Figure 4: On page 27 in the SI it is stated that the temperature-dependent UV/vis spectra have been calculated. A comparison of at least one experimentally obtained UV/vis spectrum at a certain temperature and the calculated UV/vis trace within the same graph would be nice to assess the quality of the fit.

Reply We would like to point out that we did not calculate the complete temperature-dependent UV-Vis spectra. To clarify that we specifically calculated VT-UV traces at specific wavelengths, rather than complete temperature-dependent spectra, we have included specific mentions of ‘VT-UV traces at 403 and 434 nm’ where required at page 27 and 28 of the SI.

The reviewer is right to point out that we incorporate a temperature dependent absorption coefficient of all species in the system to account for effects such as expansion of the solvent upon increasing the temperature. An example of this temperature dependent absorbance is shown in Supplementary Figure 21. During the fits, our model calculates a temperature-dependent distribution of the monomers over the various aggregates. In the fit, we use the temperature-dependent monomer distributions, the absorption coefficients determined at specific temperature

and the temperature dependency of these coefficients to construct calculated VT-UV traces at the measured wavelengths. These are then compared to the experimentally obtained VT-UV curves. For readers' information, we compared experimentally obtained UV/vis spectrum at 327 K and the one calculated in Supplementary Figure 27.

Supplementary Figure 27. (a) Calculated temperature-dependent speciation plot of a 12.5 μM solution of **6FF**. The speciation plot shows the distribution of monomers over the free monomeric, J-aggregated, and H-aggregated (Archimedean spiral) states. At 327 K, the distribution of monomers over free monomeric, J-aggregated, and H-aggregated (Archimedean spiral) states are

48, 21, and 31%, respectively. (b) Comparison of experimentally obtained spectrum at 327 K (purple open circle) and calculated absorption spectrum (black dashed line). (c) Calculated absorption spectrum (black dashed line) on the basis of the distribution of monomers at 327 K over the free monomeric (black line), J-aggregated (pink line), and H-aggregated (Archimedean spiral: green line) states.

- Figure 4a: Please explain in more detail why the apparent extinction coefficients at lower concentrations cannot be fitted with the mass balance model although the absorption spectra of the spirals and toroids are almost identical and the molecular arrangement within the spirals and toroids is similar. Is this a result of different secondary nucleation mechanisms of the spirals and toroids or is it more likely that the toroids are formed by a different aggregation mechanism?

Reply Indeed, the VT-UV data of samples with a low concentration of **6FF** could not be fitted successfully in a global fit. Although secondary nucleation events may very well play an important role in the assembly process observed in our manuscript, these events mainly influence the kinetic aspects of the polymerization. Due to microscopic reversibility, the presence or absence of secondary nucleation events will not change the system composition under thermodynamic control. Since we measured the VT-UV data at a low cooling rate, ensuring thermodynamic or close-to-thermodynamic control, different secondary nucleation events are not likely to be the cause of the inability to fit the low concentrations.

The other suggestion of the reviewer, that the toroids are formed through a different aggregation pathway, may very well be possible. At these low concentrations, the formation of the Archimedean spiral is in competition with the toroids. The pathways leading to these aggregates may be characterized by similar, but slightly different thermodynamic properties, nucleus sizes and cooperativity parameters. However, since their absorption spectra are almost identical, it is very difficult to experimentally probe the competition between these two aggregation pathways. As a result, it will be very challenging to obtain meaningful results from a computational analysis of such an energy landscape. That is, strong correlations, similar to those shown in Supplementary Figure 22 and 31 (22 and 29, respectively, in the original manuscript), between fitted parameters will likely occur if one would fit a model comprising two competitive nucleated polymerizations together with the isodesmic pathway to the data.

We have included the following sentence to make this detail clearer:

(page 12, line 16) 'Faced with this problem, we realized that changes in absorbance measured at higher (reddish plots) or lower concentrations (bluish plots) showed slightly different trends (Fig.4a), suggesting the presence of two very similar, yet subtly different aggregation pathways (*vide infra*). Since the very similar spectral properties of these aggregates prevented us to successfully fit the data, we selected appropriate data sets where only a single aggregate is observed.'

- Page 14: Please provide for 6HF also a plot of the temperature-dependent changes of the extinction coefficients at a certain wavelength as done in Figure 4a and b in the SI. Otherwise the elongation temperature of a 12.5 μM solution of 6HF of $T = 325 \text{ K}$ cannot be assessed.

Reply Thank you for pointing this out. We added the plot in Supplementary Figure 25 and 28.

Supplementary Figure 25

(a) Plots of molar absorption coefficients at 434 and 403 nm as a function of temperature: [**6FH**] = 12.5, 15.0, 17.5 μM ; -0.5 K/min . The two-pathway thermodynamic model was fitted (solid lines) to the plots indicated by the reddish filled marks. (b) Calculated temperature-dependent speciation plot of a 12.5 μM solution of **6FH**. The speciation plot shows the distribution of monomers over the free monomeric, J-aggregated, and H-aggregated (Archimedean spiral) states. (c) Matrix of scatter plots showing the optimized parameter sets with a residual $<5\%$ from the fit with lowest residual when data of 12.5, 15.0 and 17.5 μM of **6FH** are fitted. The spread of the values over a small regime indicates that a unique solution is found and *the thermodynamic parameters are accurately obtained* (Supplementary Table 4).

Supplementary Figure 28

Plots of molar absorption coefficients of **6FF** (red circle) and **6FH** (black circle) in dodecane at 403 nm as a function of temperature: $[\mathbf{6FF}] = [\mathbf{6FH}] = 12.5 \mu\text{M}$; -0.5 K/min .

- Please comment: Is the toroid structure also a double layer structure as the spiral?

Reply We added the following comment.

(page 16, line 2) Again, the separation distance between the toroids determined by AFM was larger than the length between the two fluorinated side chains in **6FF** (3.5 nm), but ADF-STEM determined the true separation distance between the consecutive toroids to be 3.2 nm (Fig. 5c, Supplementary Fig. 29).

- Figure 6a: This cartoon suggests that the primary toroid is the most inner one, i.e. that growth takes place only at the outside. However, my impression is that also filling on the inner side takes place (see Figure 6e to 6g). Please explain.

Reply Reviewer is correct. We added following sentence in the figure legend.

(Figure 6 legend) Note that new toroids are formed not only at the outer circumference but also at the inner circumference of the primary toroid, as suggested by the decrease in size of the hole observed in the AFM images.

Supporting Information

- Integrals should be added in the NMR spectra

Reply We added the integral values in the NMR spectra.

- In the reaction scheme, the dodecyloxy chain should be altered to “H25C12O” instead of “C12H25O” to make the connectivity clearer.

Reply We corrected the structure.

Thank you very much again for your comments which improved the readability of this manuscript.

Reviewer #3 (Remarks to the Author):

Sugiyasu and Takeuchi use a Zn-porphyrin with fluorinated and non-fluorinated side-chains to investigate unique double stranded Archimedean spirals and concentric toroids. The level of control is extremely impressive, and the complexity of the assembly process that arises from these partially known molecular designs is unpredictable. This is especially true for the concentration dependent morphology and assembly mechanisms. The paper will appeal to the supramolecular, organic materials and soft matter communities and thus have a big impact. The AFM microscopy, in combination with SAXS analysis are impeccable. The manuscript is further supported by a complementary modelling studies from the Eindhoven lab, which is essential to undermine the mechanistic aspect. The strong dependency on the solvent, further supports the critical role of 'fluorophilic' effect. I support publication as is.

Reply Thank you so much for your highly positive comments. We hope that this in-depth study will contribute to the research fields where self-assembly plays an important role.

REVIEWERS' COMMENTS:

Reviewer #1 (Remarks to the Author):

The authors have made significant improvements. I would like recommend the publication of the manuscript in Nature Communications. An highly related reference, Nature Commun. 2019, 10, 1399, is recommended to be cited.

Reviewer #2 (Remarks to the Author):

The authors have addressed the points raised by the reviewers. I recommend publication without further changes.

Reviewer #1 (Remarks to the Author):

The authors have made significant improvements. I would like recommend the publication of the manuscript in Nature Communications. An highly related reference, Nature Commun. 2019, 10, 1399, is recommended to be cited.

Reply

We read the suggested paper carefully; however, we did not find much relevance to our paper other than that both report self-assembly of porphyrin molecules. Therefore, we decided not to cite the suggested paper in this case. The suggested paper is actually relevant to the work we are now drafting, and we cite it in the next paper. Thank you for your suggestion.

Thank you very much again for your constructive comments with which we could improve the manuscript.

Reviewer #2 (Remarks to the Author):

The authors have addressed the points raised by the reviewers. I recommend publication without further changes.

Reply

Thank you very much again for your constructive comments with which we could improve the manuscript.